# Perspectives and experiences of the first geriatricians trained in Canada

Eric Kai-Chung Wong[ID][1,2,3]*, Alexander Day[1,2], Maria Zorzitto[1,2], Joanna E. M. Sale[3,4]

1 Division of Geriatric Medicine, St. Michael's Hospital, Unity Health Toronto, Toronto, Ontario, Canada,
2 Division of Geriatric Medicine, Department of Medicine, Temerty Faculty of Medicine, University of Toronto, Toronto, Ontario, Canada, 3 Institute of Health Policy, Management & Evaluation, University of Toronto, Toronto, Ontario, Canada, 4 Musculoskeletal Health and Outcomes Research, Li Ka Shing Knowledge Institute, St. Michael's Hospital, Unity Health Toronto, Toronto, Ontario, Canada

* eric.wong@utoronto.ca

## Abstract

Many Canadian-trained geriatricians from the subspecialty's first decade of existence continue to practice today. The objective of this study was to examine the experiences and perspectives of the earliest cohort of geriatricians in Canada. Using qualitative description method, we conducted semi-structured interviews to explore participants' experiences in training and practice. We included geriatricians who trained in Canada between 1980–1989 and were in active clinical practice as of October 2021. Each transcript was coded independently by two investigators. Thematic analysis was used to develop key themes. Fourteen participants (43% female, mean years in practice 35.9) described their choice to enter geriatric medicine, their training process, the roles of a geriatrician, challenges facing the profession and advice for trainees. Two themes were developed from the data: (i) advocacy for the older adult and (ii) geriatrics as "the road less taken". Advocacy was described as the "core mission" of a geriatrician. Participants discussed the importance of advocacy in clinical practice, education, research and disseminating geriatric principles in the health system and society. "The road less taken" reflected the challenges participants faced during training, which led to relatively few geriatricians for the growing number of older adults in Canada. Despite these challenges, participants described rewarding careers and encouraged trainees to consider the profession.

## Introduction

Geriatric medicine dates back to the pioneering work by Dr. Marjory Warren, a British physician who was assigned to look after a chronic care ward in London in 1935 [1]. Warren created an interdisciplinary, patient-centred and accessible ward with the goal of improving function in older adults with various illnesses [1]. Her vision of innovative geriatric care spread to the United States and Canada through visiting lectures. Canada first established formal geriatric training programs in the provinces of Ontario, Manitoba, and Quebec in the 1970–1980s. Geriatric medicine is currently a subspecialty of internal medicine with the Royal College of Physicians and Surgeons of Canada subspecialty certification started in 1981 [2].

**Data Availability Statement:** Data cannot be shared publicly because of personal information from interviews. However, we would be able to provide data to support the manuscript based on reasonable requests. For data requests, authors

could contact the Research Ethics Board at Unity Health Toronto: Sharon Freitag Senior Director, Research Ethics Unity Health Toronto Phone: 416.864.6060 ext. 42385 Email: sharon. freitag@unityhealth.to.

**Funding:** The Julius and Marie Molinaro Elder Care Education Fund. The funders had no role in study design, data collection and analysis, decision to publish, or preparation of the manuscript.

**Competing interests:** The authors have declared that no competing interests exist.

The creation of geriatric medicine as a subspecialty was initially met with objection in England [3]. Physicians trained in internal and family medicine felt that geriatric medicine had little to contribute to the practice of medicine, as caring for older adults was seen as an integral part of general medical practice [4]. In Canada, the field faced similar obstacles in the late 1970s as the subspecialty was created [2]. Today, official training in geriatric medicine exists in over 36 countries around the world [5] and geriatricians play roles in clinical, research, education and administrative domains. Furthermore, during the COVID-19 pandemic, geriatricians offered timely advice at a population level, advocating for resources for long-term care residents who were affected by the disease [6].

The evolution of geriatric medicine in Canada since the 1980s reflects a global movement towards recognition of aging and the importance of advancing compassionate and high-quality care for older adults. Despite this, the perspectives of geriatrician pioneers have rarely been reported in a qualitative study. The stories and experiences of geriatricians who witnessed this change offer valuable insight to inform future geriatric practice and education.

The objective of our study was to examine the perspectives and experiences of the first cohort of geriatricians trained in Canada who were still practicing as of October 2021. Their perspectives could inform (i) the progress of Canadian geriatric training programs, (ii) future education programs for trainees, and (iii) how geriatric medicine practice can evolve in the future.

## Methods

This qualitative study was conducted from December 2021 to May 2022. We invited geriatricians who were (i) trained between 1980–1989 in Canada and (ii) in full- or part-time clinical practice as of October 2021. Participants were identified by consultation with senior geriatricians at the University of Toronto, supplemented by searches of the provincial physician licensing organizations (e.g., College of Physicians and Surgeons of Ontario). To assist with recruitment across the country, we asked senior geriatric medicine faculty at the University of Toronto to facilitate invitations. We asked each participant for further suggestions (snowball sampling [7]). Informed consent was obtained in writing from each participant and interviews were conducted by phone or video conference. All interviews were conducted by one investigator (AD), who was trained by a qualitative expert (JS). Audio recordings were transcribed. Participants were assigned a numeric code and names of people were removed from the transcripts. The protocol was approved by the Unity Health Toronto Research Ethics Board (21–175). This manuscript adheres to Standards for Reporting Qualitative Research guidelines [8].

### Interview guide

An interview guide was followed for all interviews (see S1 File). The roles of the investigators and participants, as outlined by the guide, was provided to participants as a preamble. The questions were developed by the investigators and focused on the participant's experience and perspectives regarding: (i) choice of geriatric medicine, (ii) training, (iii) clinical practice, (iv) role of a geriatrician, (v) challenges and opportunities for the future and (vi) advice for trainees.

### Sample size

Based on estimates by senior geriatricians at the University of Toronto, there were likely 20–30 geriatricians across the country who met the eligibility criteria. Based on a 50% response rate [9], we planned for a sample size of 10–15 participants. Our recruitment strategy aimed to

reflect diversity in sex and region of practice of the participants. We interviewed all geriatricians who responded to our invitation.

## Analysis

After the transcripts were checked for errors, the data were coded and organized in NVivo 11, a qualitative research software (QSR International Pty Ltd, Australia). We relied on qualitative description [10] for the project design because we wanted to generate a rich and direct description of participant experiences that would inform policy [11, 12]. Thematic analysis was used to develop themes from the codes [13]. The analysis started with members of the team familiarizing themselves with the contents of the transcripts after a first read. Following that, transcripts were coded independently by two investigators (AD and EW); consensus was reached on the coding template after reviewing the first three transcripts. Codes were categorized based on the following topics: (i) reasons for choosing geriatric medicine, (ii) the training process, (iii) the role of a geriatrician in the health system, (iv) the role of a geriatrician in clinical practice, (v) challenges and opportunities for the field, and (vi) advice for geriatric trainees.

Methodologic rigour [14] was promoted by using multiple analysts [15], direct quotations to support key findings (quotes were limited to protect the identity of this group) [16], exploring analytic directions [17] and reflexivity [18, 19]. Through reflexivity, we explored personal, interpersonal, methodological and contextual factors in relation to the research goals while conducting the analysis [18, 19]. Reflexivity was done through journaling and collaboration among the investigators [18]. A qualitative research expert guided the analysis (JS). Through iterative analysis and repeated readings of the transcripts, themes were developed that represented the data.

## Results

Of the 28 eligible geriatricians contacted (50% were female), 14 geriatricians participated in the interviews (43% were female; mean years in practice was 35.9) from across Canada (8 from Ontario, 2 from Alberta, 2 from British Columbia, 1 from Quebec, and 1 from Nova Scotia). We did not collect information on gender, ethnicity and other characteristics because the population was small, potentially making the participants identifiable.

A summary of the responses to the interview questions is presented in Table 1 with example quotes for each response. Participants discussed their reasons for choosing geriatric medicine and their training process. They described high-level roles of a geriatrician in the health system and specific roles when conducting a geriatric assessment. Participants also outlined challenges and opportunities for the profession, and offered advice for geriatric trainees. In particular, participants encouraged trainees to explore clinical and non-clinical opportunities to develop a sustainable career path. They also advised trainees to join a group practice for mutual support and to avoid burnout.

There were two major themes that reflected participants' discussions: (i) advocacy for the older adult and (ii) geriatrics as "the road less taken".

### Theme 1: Advocacy for the older adult

Advocacy was mentioned by multiple participants as the key role of a geriatrician. A participant recalled practices in the past where older adults were excluded from treatments that were felt to be of little benefit. Participant 11 recounted, "We tried to be an advocate for the elderly. That was our biggest role. When streptokinase came about to thrombolize acute myocardial infarction. . . the hospital said we're not going go to waste this expensive drug on old people. We had to fight with the cardiologists." Advocating for older adults and countering ageism

**Table 1. Response summary of the interview questions with example quotes.**

| Question topic | Main responses | Example quotes |
|---|---|---|
| **1. Reason for choosing geriatric medicine** | • Positive experience during training | "I think the initial attraction was more intellectual, dealing with complexity, looking at problems in a different light and maybe framing problems in a different way"–Participant 9 |
| | • Holistic and comprehensive care | "I realized that if anything was holistic in terms of one's approach to health, it was geriatrics"–Participant 6 |
| | • Grandparents | "I was really close to my maternal grandmother. I had a great relationship with her and always loved listening to her stories."–Participant 4 |
| | • Mentorship | [Multiple stories of specific mentors] |
| | • Reflections of choice | "It turned out to be spectacularly good luck. I've had a spectacularly rewarding and interesting and wonderful career. It was one of my better decisions in life."–Participant 12 |
| **2. The training process** | • Training outside of Canada | "Geriatrics was relatively new in Canada at that time. [The program] was supportive of me going somewhere else to see [geriatrics services] set up in a different way with the hopes that I would bring that back to Canada, which I eventually did."–Participant 2 |
| | • Self-directed learning | "The training in geriatrics was a self-learning process. The formal training was in internal medicine."–Participant 14 |
| | • Lack of structure | "It was much less standardized. . . more of an apprenticeship model."–Participant 10 |
| | • Challenges for female trainees | "I think there were inequities for women that were more apparent back then. . . I think some of the male physicians were maybe able to better advocate for. . . more resources. There were no programs like maternity leave or childcare support."–Participant 2 |
| | • Gaps in current training programs | "I think the biggest gap is how to set up a practice. It depends where you are going to practice, but if you are going to community, it takes a lot to carve out the resources that you need. . . And I'm not sure how well we provide trainees with the [knowledge] to do that."–Participant 3 "There are those of us who come to medicine with a different background. I wonder if [we are] losing out on applicants who've had challenges in their life, and I'm not just talking financial [challenges], but academic or otherwise who actually would make amazing physicians, but they don't get the opportunity because their CV just doesn't match that template. . . That's sometimes how I feel when I look at the applicants: I see very accomplished individuals. . . but I'm not sure how much diversity we are attracting."–Participant 6 |
| **3. Role of a geriatrician in the health system** | • Advocacy | "Maybe that needs to be considered as one of the qualities of a geriatrician–to be an outspoken advocate."–Participant 14 |
| | • Education | "Our job is to educate people so that they can use geriatric principles"–Participant 13 |
| | • Clinical care | "[The] number one [role] is providing consultation and education around complex problems in old people–in other words, the clinical role."–participant 3 |
| | • Research | Another role of geriatrics is "the pursuit of new knowledge. And I think that that's something that has grown enormously in recent years."–participant 3 |
| | • Health policy and systems change | "I think our role will be to educate policy makers and the public and by providing models of care that work."–Participant 6 |
| **4. Role of a geriatrician in clinical practice** | • Holistic and comprehensive | "I think we have a more holistic approach and we take the time to know the family and the social circumstances."–Participant 4 |
| | • Patient-centred care | "We don't characterize individuals by their pathologies. We characterize them based on their frailty and what matters to them."–Participant 13 |
| | • Understanding complexity | "We are good at dealing with undefined and nonspecific problems that people don't know what to do with."–Participant 12 |
| | • Patience | "I think that geriatricians have the requisite humility to imagine they can't get it all sorted out the first time they see someone, whereas many other specialties are so used to pattern recognition that they imagine they can forecast the entire course of a patient from the first time they see them."–Participant 10 |
| | • Working in teams | "We know how to work as a team and we're happy with it."–Participant 10 |
| | • Implementation of recommendations | "The published data shows a consultant that comes in and leaves accomplishes almost nothing. A consultant that comes in, follows up and implements the recommendations actually works."–Participant 5 |
| | • Focus on frailty rather than age cutoff | "So do we define our work by how old someone was or by the geriatric giants?"–Participant 13. |

*(Continued)*

**Table 1.** (Continued)

| Question topic | Main responses | Example quotes |
|---|---|---|
| **5. Challenges and opportunities for the future** | • Insufficient number of geriatricians in Canada | "I don't think there will ever in my, or in your lifetime, be enough geriatricians to see all the older people."–Participant 1 |
| | • Disease-specific focus versus comprehensive care | Geriatrics is "moving away from home visits and towards dementia. There is a move to constrain what geriatric medicine does."–Participant 7 |
| | • Focusing on acute vs. outpatient care | "In the 80s, there was more emphasis and interest in community-based practice. My sense now is that we've retreated back into hospitals."–Participant 9 |
| | • Opportunities for the future* | "I think we're going to see that expansion of geriatricians being more available in non-academic settings."–Participant 13 |
| **6. Advice for trainees** | • Choosing a career path they enjoy | The most rewarding career can be found at the intersection of "what you love, what the world needs, what you are good at and what you will be paid for."–Participant 4 describing the concept of ikigai<br>"We all come with different talents, different strengths, different ideas of what our purpose is. My advice is for individuals to explore that and understand what it is that makes them happy, as a geriatrician or just as a person."–Participant 13 |
| | • Develop clinical and non-clinical skills | "I would encourage [trainees] to not distance themselves too far from acute care, because that really keeps you up on your internal medicine."–Participant 8 |
| | • Finding opportunities outside of clinical medicine | "You have to be interested in some administrative role as well because sometimes you have to have power in order to get changes to happen."–Participant 14 |
| | • Develop teams and strong work relationships | "What's really important for young doctors is to find a group to join. Don't practice on your own. You need to practice in a group to . . . share experiences and get support from each other."–Participant 11 |

*Additional responses for future opportunities for geriatric medicine: (i) helping older adults age at home, (ii) advancing dementia therapy, (iii) expanding rural geriatric care, (iv) developing geriatric subspecialty programs (e.g., geriatric cardiology, geriatric nephrology), (v) translating research into practice, (vi) making hospitals and long-term care homes more geriatric friendly, (vii) personalized medicine

was felt to be the "core mission" of geriatric medicine (participant 12). The levels of advocacy ranged from that of the individual patient level to that of the health care system and societal in general. The hazards that older adults faced in long-term care and other settings during the COVID-19 pandemic was brought up as an example where geriatricians influenced care priorities broadly. Advocacy in health policy was supported by "the rise in prominence" of geriatrician researchers over the decades (participant 3). Participant 14 stated, "maybe that needs to be considered as one of the qualities of a geriatrician–to be an outspoken advocate."

Features of a geriatric assessment reflected advocacy in everyday practice. Participants outlined the several key characteristics that a geriatric assessment should have, including being comprehensive or holistic, being patient-centred, having patience, and understanding complexity. Being holistic and comprehensive were characteristics that differentiated geriatricians from other specialties, and several participants cited this as a reason for choosing to enter the profession. Identifying missed issues (e.g., functional decline, falls, or mood issues), incorporating the family/caregiver perspective, and addressing the determinants of health were important elements of a geriatric assessment. Participants also preferred a balanced approach of using evidence in the context of the patient's frailty and preferences. Participant 13 reflected on this approach that prioritized patient preferences: "People were very fortunate if they got to see a geriatrician. Part of that is the fact that we don't characterize individuals by their pathologies. We characterize them based on their frailty and what matters to them."

Navigating medical complexity and clinical uncertainty were additional clinical roles used for advocacy. Geriatric care was contrasted with the principle of Occam's razor, which is often taught in medical schools. Medical students learn that when patients present with multiple symptoms, they should find a single unifying diagnosis that explains all of them. Older adults

often do not have a single unifying diagnosis that will address their problems. Participants noted that a geriatrician often identifies multiple medical problems from a complex presentation, which have to be prioritized and addressed separately. Participants also felt that strengths of geriatricians included appreciating clinical uncertainty and nonspecific problems. Participant 12 reflected, "We are good at dealing with undefined and nonspecific problems that people don't know what to do with."

The setting where geriatricians could best advocate and care for patients was reflected in divergent views. Some participants suggested that geriatricians should practice in acute care settings, particularly on acute medical or surgical units. Maintaining the internal medicine knowledge base and finding opportunities to interact with trainees were cited as reasons why acute medicine was important. Engaging trainees could generate interest in the subspecialty and instil geriatric principles in training. On the contrary, some participants emphasized home visits and community-based practices as essential to a geriatrician's role because early management of geriatric issues might prevent hospitalizations. As participant 9 said, "the best hospital stay for an older person is to avoid it." Participants also noted gaps in the training programs included limited community practice exposure and administrative management skills. As a corollary, participants wanted geriatricians to play a more prominent role in transforming hospitals and cities in a more geriatric friendly way. "I think we haven't properly evaluated the evidence on how hospitals and long-term care facilities should be built to make them truly senior friendly. . . when a facility is senior friendly, it's friendly to everybody who's there" (participant 1).

Education was another strategy participants relied upon to advocate for older adults. Participants noted that education about geriatric medicine should start in medical school and be widely disseminated to generalist and specialist physicians ("Geriatricizing other specialties"). Educating the general public about healthy aging was seen as an important step to change ageist views. Participants also had mixed thoughts about whether geriatric medicine should focus on dementia care or pursue a broader mission to care comprehensively for older adults. Some participants saw dementia biomarkers and targeted therapy as key to the field's future, while others wanted the field to be grounded in advocacy (and not constrained by focusing on dementia) and integrated care for older adults. Geriatrics was viewed as "moving away from home visits and towards dementia. There is a move to constrain what geriatric medicine does" (participant 7). Another participant (participant 9) felt that frailty was a negative concept and wondered whether the specialty would be better served by framing aging in a positive light: "I'm not sure that the current rapture with frailty is a good thing. Frailty is a very negative concept and I always felt that we might be better served if we considered the positives and strengths of an older individual rather than looking at their deficits and weaknesses."

## Theme 2: Geriatrics was considered "the road less taken"

The second theme represented the challenges participants faced through their career as a geriatrician. Choosing the road less taken, participants reported overcoming barriers in choosing the subspecialty and training in an unstructured program. These challenges resulted in a health system with an insufficient number of geriatricians for the aging population. However, attitudes towards geriatric medicine improved during these participants' careers, and many felt it was a rewarding choice.

In their choice to enter the field, participants appeared to favour geriatric medicine because it was "more intellectual" and challenging compared to other internal medicine subspecialties. Some participants reported being discouraged from choosing geriatrics by other internal medicine specialists because it was seen as a subspecialty with no future. Most participants recalled

having mentorship early in training as a key factor in their choice to train in geriatric medicine. Mentors included senior geriatricians trained outside of Canada and physicians from other specialties. Participant 5 recalled, "[On] my first day of medical school, the Dean. . . said, 'Unless you go into pediatrics or obstetrics, you're going to be treating the elderly, [so] you should train in geriatrics.'" Participants generally felt that attitudes towards geriatrics had improved in later years. Improvement in compensation also encouraged more trainees to consider the subspecialty.

Several participants underwent additional geriatric training outside of Canada due to limited Canadian training programs at the time. Most participants who trained abroad chose to go to the United Kingdom and the United States, as programs in those countries were more established. Participants reported that training programs in Canada during the 1980s lacked structure; the training was primarily done by apprenticeship and the learning was mainly self-directed. Participants recalled reading after hours, organizing seminars with other residents, and going to conferences to learn geriatric medicine. Some participants conducted research projects, but reflected on the lack of education on how to set up and evaluate a geriatric program after graduation. Participant 6 told us she had wanted formal leadership training, but this was not available at the time. Participant 13 experienced challenges during training due to the lack of respect for geriatric trainees: "The first three months of my residency were not good. That was because of total lack of respect for geriatrics and what we could do. I just remember seeing a consult and another consultant would walk in and say 'Oh, you're geriatrics.' And then just totally ignore me." The participant later moved to another training site where the experience was much more supportive. Female trainees also faced barriers in advocating for resources and child care support. Participants reported that these barriers had improved over time. Overall, current geriatric training programs were viewed as more organized and formalized.

Reflecting on their own career choices, participants offered some advice to trainees. Several participants encouraged trainees to reflect on their strengths and weaknesses and pursue a career that they could enjoy. A participant used the Japanese concept of ikigai ("reason for being" [20]) to guide a trainee's search for an enjoyable career. The most rewarding career could be found at the intersection of "what you love, what the world needs, what you are good at and what you will be paid for" (participant 4). The concept could be used to find something within geriatric medicine to focus on as well. "We all come with different talents, different strengths, different ideas of what our purpose is. My advice is for individuals to explore that and understand what it is that makes them happy, as a geriatrician or just as a person" (participant 13).

Participants acknowledged the relatively small number of geriatricians compared with the growing number of older adults. This was related to their preference to stay in the academic setting, so that more trainees could be encouraged to choose this subspecialty. Some participants were not aware of the aging demographic trends at the beginning of training, so they reported that their career was more rewarding than expected. For example, one participant told us, "It turned out to be spectacularly good luck. I've had a spectacularly rewarding and interesting and wonderful career. It was one of my better decisions in life" (participant 12).

## Discussion

The results of our study encapsulated the experiences and reflections of the first geriatricians trained in Canada. The themes illustrated a winding path taken by early geriatricians to train and practice in this field. Participants generally reported an improvement in the way geriatric medicine is perceived and the way older adults are treated. They also presented a vision of the

role geriatricians should play in the health system and at the societal level. This type of narrative reflection comparing the past and present of geriatric medicine is seldom reported in the literature. The findings can inform the design of future geriatric medicine training programs to better equip future geriatricians for diverse roles in the health care system.

The theme of advocacy against ageism appeared to underlie many of the responses by participants. From their choice to train in geriatric medicine to their reflections on the most important role of a geriatrician, participants demonstrated the power of geriatric medicine to improve the lives of older adults. Ageism was not only apparent in the care of patients, but it was also evident in the way the profession was viewed. Although participants in our study witnessed a significant improvement in the respect of geriatric medicine, trainees still cite "lack of prestige" as a common reason for not choosing the specialty [21, 22]. Perhaps this is the reason why advocacy was seen as the core mission of geriatric medicine in this group of geriatricians. Existing studies have outlined the ways a population can reduce ageism, including education and intergenerational contact [23]. However, studies investigating the role of geriatricians in reducing ageism are lacking. Future studies can explore methods of advocacy for geriatricians to implement in practice, education, research and policy to effectively reduce ageism.

Participants had contrasting views on where the subspecialty should focus in the future. Defining the target population has been debated by geriatricians in recent years [24, 25]. There is an insufficient number of geriatricians for the growing needs of an aging population. If geriatricians cannot directly care for every older adult with frailty or geriatric syndromes, then disseminating core geriatric principles into institutions, healthcare systems and even the broader society is needed [26]. Our study participants agreed with this goal. Furthermore, the focus on frailty and deficits in geriatric medicine was felt to distort a positive view of aging. Recent publications discussed the importance of marketing geriatric medicine to the public [6], and reframing the specialty with a more balanced view of aging [27]. If defining patients by their diseases is counter to the philosophy of geriatric medicine today, then perhaps defining patients by their deficits will also be outdated. Future research should determine how we can best integrate a positive view on aging while addressing frailty or geriatric syndromes.

Reflecting on their career, we asked participants for their advice to future geriatricians. Several participants advised graduates to work in groups and avoid working in isolation. In Canada, most geriatricians work in group or hospital-based practices and only 10% work in a solo practice [28]. Most Canadian geriatricians also work in academic hospital settings (59%) [28], so some of our participants felt that there is a need to increase community- and rural-based geriatricians. Regardless of the practice setting, participants were encouraged to develop skills outside of clinical medicine to become leaders for the health system. In our questions, we did not distinguish the level of health system (e.g., local institution, provincial, national or international) that a geriatrician should influence. However, the themes expressed indicate a broader vision for how future geriatricians should shape society's view of aging.

The strengths of this study included the use of multiple analysts, the use of direct quotations to support our key findings, and the engagement of a team that includes geriatricians and clinical epidemiologists with expertise in qualitative research. Limitations of this study include the small sample of geriatricians interviewed (but consistent with the sample size of qualitative studies [29]), and a relatively low proportion of female participants, which is not reflective of today's profile of the specialty [28]. Based on our eligibility criteria, we also possibly missed recruiting geriatricians who had retired recently or who were not known to faculty at the University of Toronto. We also did not include younger geriatricians to determine if their experiences are different, a topic which could be addressed in future studies.

## Conclusion

Interviews with the first group of Canadian-trained geriatricians revealed important changes in training, professional practice and patient characteristics through the years. Advocating for older adults was consistently stated as the most important role of a geriatrician. The diverse roles of geriatricians and growing needs of the population offer numerous opportunities to shape the healthcare system in a geriatric-friendly way.

## Supporting information

**S1 File. Interview guide.**
(PDF)

## Acknowledgments

We thank all of the participating geriatricians for taking the time to share their experiences and thoughts for the profession. We thank Dr. Sharon Straus for providing valuable feedback and reviewing the manuscript. We also thank Dr. Barbara Liu, Dr. Dov Gandell, Dr. Camilla Wong, and Dr. Heather Gilley for assistance in connecting with available geriatricians.

## Author Contributions

**Conceptualization:** Eric Kai-Chung Wong, Alexander Day, Maria Zorzitto, Joanna E. M. Sale.

**Data curation:** Alexander Day, Joanna E. M. Sale.

**Formal analysis:** Eric Kai-Chung Wong, Alexander Day, Maria Zorzitto, Joanna E. M. Sale.

**Funding acquisition:** Maria Zorzitto.

**Investigation:** Eric Kai-Chung Wong, Alexander Day, Joanna E. M. Sale.

**Methodology:** Eric Kai-Chung Wong, Alexander Day, Maria Zorzitto, Joanna E. M. Sale.

**Project administration:** Maria Zorzitto, Joanna E. M. Sale.

**Resources:** Maria Zorzitto, Joanna E. M. Sale.

**Software:** Alexander Day, Joanna E. M. Sale.

**Supervision:** Eric Kai-Chung Wong, Maria Zorzitto, Joanna E. M. Sale.

**Validation:** Eric Kai-Chung Wong, Alexander Day, Maria Zorzitto, Joanna E. M. Sale.

**Writing – original draft:** Eric Kai-Chung Wong, Alexander Day.

**Writing – review & editing:** Eric Kai-Chung Wong, Maria Zorzitto, Joanna E. M. Sale.

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
