## [Decision Letter · Decision Letter 0]

10 May 2023

PONE-D-23-05523

Perspectives and experiences of the first geriatricians trained in Canada

PLOS ONE

Dear Dr. Wong,

Thank you for submitting your manuscript to PLOS ONE. After careful consideration, we feel that it has merit but does not fully meet PLOS ONE’s publication criteria as it currently stands. Therefore, we invite you to submit a revised version of the manuscript that addresses the points raised during the review process.

We look forward to receiving your revised manuscript.

Kind regards,

Mario Ulises Pérez-Zepeda, M.D., Ph.D.

Academic Editor

PLOS ONE

Journal Requirements:

Additional Editor Comments:

Dear authors, thank you for this interesting work. Reviewers have minor concerns, and I will be happy to consider a revised version of your work for publication.

Reviewers' comments:

Reviewer's Responses to Questions

**Comments to the Author**

1. Is the manuscript technically sound, and do the data support the conclusions?

Reviewer #1: Yes

Reviewer #2: Yes

2. Has the statistical analysis been performed appropriately and rigorously? 

Reviewer #1: I Don't Know

Reviewer #2: N/A

3. Have the authors made all data underlying the findings in their manuscript fully available?

Reviewer #1: No

Reviewer #2: Yes

4. Is the manuscript presented in an intelligible fashion and written in standard English?

Reviewer #1: Yes

Reviewer #2: Yes

5. Review Comments to the Author

Reviewer #1: This manuscript- despite being an easy read -addresses a very important perspective to young physicians considering a geriatric specialty. It also may help physicians thinking about establishing a geriatric specialty in their countries (Geriatrics is not yet present in many countries).

Authors stated that data are not fully disclosed and are not being shared publicly. Is it possible to share after removing any personal details?

The manuscript is well-written. Apart from a typo in table 1 section about advice for trainees- distance them themselves, there are no other remarks.

Reviewing appendix 1: the questionnaire included a question about what the participants found something missing from the training of current geriatric trainees, this was not mentioned in the manuscript and could provide meaningful insight to anyone involved in the establishment of training programs for geriatricians.

Reviewer #2: This a very interesting study aiming to examine the perspectives and experiences of the first cohorts of geriatricians in Canada. Through structured interviews experienced geriatricians told their impressions of the specialty. The main topics discussed were the role of advocacy for older people and geriatrics as the less taken road.

I found it very compelling and interesting, I only have some minor comments.

In the introduction, you could complement it by adding evidence on the subject from Canada or elsewhere. Is there any information on perspectives from geriatricians? Show there is a gap of knowledge, and why this would be relevant to know.

In the discussion, it might be worth it to compare the reasons to choose geriatrics mentioned by the respondents with younger geriatricians. Also is there any evidence that geriatricians can decrease ageism?

6. PLOS authors have the option to publish the peer review history of their article (what does this mean?). If published, this will include your full peer review and any attached files.

Reviewer #1: **Yes: **Samia A. Abdul-Rahman

Reviewer #2: No

---

## [Author Response · Author response to Decision Letter 0]

29 May 2023

Comments Author’s responses

Editor’s comments 

1. Please ensure that your manuscript meets PLOS ONE's style requirements, including those for file naming. The manuscript has been formatted to meet PLOS ONE’s requirements. We could not find instructions for file naming – please let us know if the file names are incorrect and we will correct them. 

 Our study included practicing physicians; no minors were included. We have added detail about the consent in Methods section. 

“Informed consent was obtained in writing from each participant and interviews were conducted by phone or video conference.” – page 4

 We have corrected the funding information in both locations to:

“The Julius and Marie Molinaro Elder Care Education Fund”

 We have carefully considered the data sharing requirement. The main issue is that the participants (Canada’s first geriatricians) is a very small group. Their accounts and experiences are relatively unique, making the transcripts readily identifiable by certain audiences, even with deidentification. Some of the accounts are personal, so we have not received consent for release of the transcripts. However, we would be able to provide data to support the manuscript based on reasonable requests. For data requests, authors could contact the Research Ethics Board at Unity Health Toronto.

5. Please review your reference list to ensure that it is complete and correct. If you have cited papers that have been retracted, please include the rationale for doing so in the manuscript text, or remove these references and replace them with relevant current references. Any changes to the reference list should be mentioned in the rebuttal letter that accompanies your revised manuscript. If you need to cite a retracted article, indicate the article’s retracted status in the References list and also include a citation and full reference for the retraction notice. The reference list is correct and none of the articles have been retracted to our knowledge. 

Reviewer’s comments 

Reviewer #1: This manuscript- despite being an easy read -addresses a very important perspective to young physicians considering a geriatric specialty. It also may help physicians thinking about establishing a geriatric specialty in their countries (Geriatrics is not yet present in many countries). Thank you. 

Authors stated that data are not fully disclosed and are not being shared publicly. Is it possible to share after removing any personal details? We have considered this. Since the pool of original geriatricians is small, their accounts and encounters are relatively unique. Even if we removed personal identifiers, their experiences alone may allow them to be identified. We have therefore decided not to release the original transcripts, some of which contain very personal information. See data sharing response above. 

The manuscript is well-written. Apart from a typo in table 1 section about advice for trainees- distance them themselves, there are no other remarks. Thank you. We corrected the error. The line from table 1 now reads:

“I would encourage [trainees] to not distance themselves too far from acute care, because that really keeps you up on your internal medicine.” – Participant 8

Reviewing appendix 1: the questionnaire included a question about what the participants found something missing from the training of current geriatric trainees, this was not mentioned in the manuscript and could provide meaningful insight to anyone involved in the establishment of training programs for geriatricians. Added line in results:

“Participants also noted gaps in the training programs included limited community practice exposure and administrative management skills.” – page 11 

Also added 2 quotes in Table 1. 

“Gaps in current training programs” under “the training process”

“I think the biggest gap is how to set up a practice. It depends where you are going to practice, but if you are going to community, it takes a lot to carve out the resources that you need… And I’m not sure how well we provide trainees with the [knowledge] to do that.” – Participant 3

“There are those of us who come to medicine with a different background. I wonder if [we are] losing out on applicants who’ve had challenges in their life, and I’m not just talking financial [challenges], but academic or otherwise who actually would make amazing physicians, but they don’t get the opportunity because their CV just doesn’t match that template… That’s sometimes how I feel when I look at the applicants: I see very accomplished individuals… but I’m not sure how much diversity we are attracting.” – Participant 6

Reviewer #2: This a very interesting study aiming to examine the perspectives and experiences of the first cohorts of geriatricians in Canada. Through structured interviews experienced geriatricians told their impressions of the specialty. The main topics discussed were the role of advocacy for older people and geriatrics as the less taken road. 

In the introduction, you could complement it by adding evidence on the subject from Canada or elsewhere. Is there any information on perspectives from geriatricians? Show there is a gap of knowledge, and why this would be relevant to know. We have added the following line to introduction:

“Despite this, the perspectives of geriatrician pioneers have rarely been reported in a qualitative study.” – page 3

We could not find any similar studies interviewing geriatricians directly on their career and perspectives. 

In the discussion, it might be worth it to compare the reasons to choose geriatrics mentioned by the respondents with younger geriatricians. Also is there any evidence that geriatricians can decrease ageism? We could not find data to support a geriatrician’s role in reducing ageism and we agree this is an important future research direction. We also agree with the point about seeking perspectives from younger geriatricians. 

We have added the following to the manuscript:

“However, studies investigating the role of geriatricians in reducing ageism are lacking.” Page 15

Under limitations, we have added:

“We also did not include younger geriatricians to determine if their experiences are different, a topic which could be addressed in future studies.” – Page 16

---

## [Editor Report · Decision Letter 1]

14 Jun 2023

Perspectives and experiences of the first geriatricians trained in Canada

PONE-D-23-05523R1

Dear Dr. Wong,

We’re pleased to inform you that your manuscript has been judged scientifically suitable for publication and will be formally accepted for publication once it meets all outstanding technical requirements.

Kind regards,

Mario Ulises Pérez-Zepeda, M.D., Ph.D.

Academic Editor

PLOS ONE

Additional Editor Comments (optional):

Dear Authors, thank you very much for your revised version. I personally enjoyed very much reading this work; in addition, you have addressed the reviewers' suggestions and I am happy to move forward with your work. I would love to see more works like this, since they are urgently needed in geriatrics.
---

## [Editor Report · Acceptance letter]

26 Jun 2023

PONE-D-23-05523R1 

Perspectives and experiences of the first geriatricians trained in Canada 

Dear Dr. Wong:

I'm pleased to inform you that your manuscript has been deemed suitable for publication in PLOS ONE. Congratulations! Your manuscript is now with our production department. 

Kind regards, 

on behalf of

Dr. Mario Ulises Pérez-Zepeda 

Academic Editor

PLOS ONE